

# Shoulder mobility and strength impairments in patients with rotator cuff related shoulder pain: a systematic review and meta analysis

Daniel Manoso-Hernando[1], Javier Bailón-Cerezo[1], Santiago Angulo-Díaz-Parreño[2], Álvaro Reina-Varona[3], Ignacio Elizagaray-García[1] and Alfonso Gil-Martínez[1,4]

[1] CranioSPain Research Group, Centro Superior de Estudios Universitarios La Salle, Madrid, Spain
[2] Facultad de Medicina, Universidad de San Pablo CEU, Boadilla del Monte, Madrid, Spain
[3] Motion in Brains Research Group, Centro Superior de Estudios Universitarios La Salle, Madrid, Spain
[4] Unidad de Fisioterapia, Hospital Universitario La Paz-Carlos III (IdiPAZ), Madrid, Spain

Corresponding author
Ignacio Elizagaray-García,
jeliga@lasallecampus.es

## ABSTRACT

**Background**. The methods previously proposed in the literature to assess patients with rotator cuff related shoulder pain, based on special orthopedic tests to precisely identify the structure causing the shoulder symptoms have been recently challenged. This opens the possibility of a different way of physical examination.

**Objective**. To analyze the differences in shoulder range of motion, strength and thoracic kyphosis between rotator cuff related shoulder pain patients and an asymptomatic group.

**Method**. The protocol of the present research was registered in the International Prospective Register of Systematic Review (PROSPERO) (registration number CRD42021258924). Database search of observational studies was conducted in MEDLINE, EMBASE, WOS and CINHAL until July 2023, which assessed shoulder or neck neuro-musculoskeletal non-invasive physical examination compared to an asymptomatic group. Two investigators assessed eligibility and study quality. The Newcastle Ottawa Scale was used to evaluate the methodology quality.

**Results**. Eight studies ($N = 604$) were selected for the quantitative analysis. Meta-analysis showed statistical differences with large effect for shoulder flexion (I2 = 91.7%, $p < 0.01$, HG = −1.30), external rotation (I2 = 83.2%, $p < 0.01$, HG = −1.16) and internal rotation range of motion (I2 = 0%, $p < 0.01$, HG = −1.32). Regarding to shoulder strength; only internal rotation strength showed statistical differences with small effect (I2 = 42.8%, $p < 0.05$, HG = −0.3).

**Conclusions**. There is moderate to strong evidence that patients with rotator cuff related shoulder pain present less shoulder flexion, internal and external rotation range of motion and less internal rotation strength than asymptomatic individuals.

## INTRODUCTION

Rotator cuff pathology is considered one of the most common causes of shoulder pain and disability, affecting millions of people worldwide (*Lewis, 2009a*; *Hinsley et al., 2022*) and causing a great impact in the basic daily activities, including activities such as eating, working or getting dressed (*Bennell et al., 2007*). Furthermore, the long-term outcome in a significant patient proportion is poor (*Lewis, 2009a*).

The term historically used to describe this condition was subacromial impingement syndrome, where shoulder pain would arise as a result of irritation onto the subacromial bursa and rotator cuff tendons from the under-surface of the over-lying anterior aspect of the acromion (*Neer, 1972*). Recent publications have challenged this diagnosis as no statistical differences have been found in subacromial space between subacromial impingement patients and asymptomatic population (*Desmeules et al., 2004*; *Kalra et al., 2010*; *Michener et al., 2015*; *Navarro-Ledesma et al., 2017*). In addition, different studies have demonstrated that subacromial decompression surgery provided no important benefit compared with placebo surgery or exercise therapy (*Henkus et al., 2009*; *Ketola et al., 2016*; *Kolk et al., 2017*; *Beard et al., 2018*; *Paavola et al., 2018*; *Lähdeoja et al., 2020*).

A series of clinical terms have been proposed to actively move away from the term subacromial impingement syndrome (*Diercks et al., 2014*; *Whittle & Buchbinder, 2015*). Rotator cuff related shoulder pain (RCRSP) is an umbrella term that encompasses a spectrum of shoulder conditions including; subacromial pain syndrome, subacromial impingement syndrome, rotator cuff tendinopathy, bursitis and symptomatic partial and full thickness rotator cuff tears (*Lewis, 2016*). It was proposed to avoid uncertainties associated with scientifically outdated diagnoses and to help the patient make sense of their experience of shoulder pain and weakness (*Lo, van Griensven & Lewis, 2022*).

Establishing a functional diagnosis is the primary objective within the professional practice of physical therapists (*Requejo-Salinas et al., 2022*). However, RCRSP etiology is not clear and the presence of tendinopathy associated signs such as degenerative lesions, rotator cuff tears, morphological changes or calcifications are widely cited in the absence of pain or functional deficit (*Hegedus et al., 2010*; *Minagawa et al., 2013*; *Russell et al., 2014*; *Sejersen et al., 2015*). In this sense, natural history studies suggest that asymptomatic rotator cuff tears are likely to become symptomatic over time (*Yamaguchi et al., 2001*; *Mall et al., 2010*; *Moosmayer et al., 2013*; *Keener et al., 2015*). Nonetheless, the extent to which an asymptomatic rotator cuff tears impacts shoulder function remains unclear (*Lawrence, Moutzouros & Bey, 2019*); however, some predictive factors to identify asymptomatic patients with shoulder pathology have been proposed in the literature (*Lawrence et al., 2024*). Additionally, physical examination based on special orthopedic tests to identify the source of shoulder pain has been challenged (*Clark, Sidles & Matsen, 1990*; *Clark & Harryman, 1992*; *Tempelhof, Rupp & Seil, 1999*; *Hegedus et al., 2008*; *Hegedus et al., 2012*; *Lewis, 2009b*; *Boettcher, Ginn & Cathers, 2009*; *May et al., 2010*). Moreover, altered scapular kinematics are believed to contribute to rotator cuff pathology by reducing the subacromial space; however, the effects of scapular dyskinesis on subacromial proximities are not clear (*Lawrence, Braman & Ludewig, 2020*) and there is currently insufficient evidence

to recommend any instrument for its assessment (*D'hondt et al., 2020*). Lastly, thoracic kyphosis seems not to be an important contributor to the development of unspecific shoulder pain (*Barrett et al., 2016*); however, no systematic review has previously specifically evaluated this variable in RCRSP population.

A recent Delphi study with nine shoulder physiotherapist experts concluded that the diagnosis of RCRSP should be based on presence of pain during shoulder active abduction (ABD) and active external rotation (ER) (*Littlewood et al., 2019*) as Lewis also described (*Lewis, 2009b*). While a functional approach is also suggested for the treatment of RCRSP, with a structural exercise program being the first line intervention (*Haahr & Andersen, 2006*; *Kukkonen et al., 2014*; *Ketola et al., 2016*), literature reflects contradictory results in regards to what kind of exercise program is more effective (*Kuhn, 2009*; *Littlewood et al., 2012*; *Littlewood, May & Walters, 2013*). One of the reasons not to reach optimal outcomes could be the fact that there is a lack of scientific research relative to the main physical impairments in RCRSP patients. This can lead to unspecific exercise programs, in contrast to what has been suggested in other musculoskeletal conditions (*O'Sullivan, 2005*; *Falla & Hodges, 2017*).

Therefore, it seems essential to increase the knowledge relative to physical impairments in this population. To the best knowledge of the authors of the present study, no previous systematic review or meta-analysis has analyzed the differences in shoulder range of motion (ROM), shoulder strength and thoracic kyphosis between RCRSP patients and an asymptomatic group (AG). This meta-analysis explored the shoulder ROM, strength and thoracic kyphosis differences between both groups during the physical examination of the shoulder to recognize which physical impairments are more common in these patients.

## MATERIALS & METHODS

### Data sources and searches

The protocol of the present research was registered in the International Prospective Register of Systematic Review (PROSPERO) (registration number CRD42021258924) (*Schiavo, 2019*) and was carried out through the PRISMA standards protocol (*Page et al., 2021*). (Detailed in File S8 PRISMA CHECKLIST).

To avoid bias, two independent reviewers (DMH and IEG) conducted the search, implementing an agreement for the search strategy and study selection. The MEDLINE, EMBASE, WOS, and CINAHL databases were assessed without language and temporal filters, with the search ending on the 21st July 2023.

The same searching equation was performed by both reviewers: (((((((rotator cuff[MeSH Terms]) OR (subacromial impingement syndrome[MeSH Terms])) OR (shoulder pain[MeSH Terms])) OR (bursitis[MeSH Terms])) OR (tendinopathy[MeSH Terms])) OR ((((((shoulder tend*) OR (shoulder burs*)) OR (supraspinatus)) OR (infraspinatus)) OR (teres minor)) OR (subscapularis))) AND ((((((((((physical examination[MeSH Terms]) OR (range of motion[MeSH Terms])) OR ("pain"[MeSH Terms])) OR (pain threshold[MeSH Terms])) OR (myofascial trigger point pain[MeSH Terms])) OR (muscle strength[MeSH Terms])) OR (dynamometer, muscle strength[MeSH Terms])) OR

(((((("clinical test") OR ("pain location"))) OR ("physical examination test")) OR (scapular dyskin*))) OR ((electromyography[MeSH Terms]) OR ("surface electromyography")))) AND ((("shoulder"[MeSH Terms]) OR ("spine"[MeSH Terms])) OR ((((neck) OR (cervical)) OR ("thoracic spine")) OR ("shoulder girdle"))) Filters: Clinical Study, Comparative Study, Evaluation Study, Observational Study, Validation Study, Humans Sort by: Most Recent.

Likewise, the same descriptors were adapted and combined to perform the same searching in EMBASE, WOS, and CINHAL databases. In addition, reference lists of included studies were reviewed to identify additional studies, those articles that fulfilled the inclusion and exclusion criteria were included and progressed to the next phase (Records identified from citation searching).

### Study selection

To be included in this meta-analysis, the studies had to have an observational design in which a shoulder or neck neuro-musculoskeletal non-invasive physical examination (active or passive shoulder ROM, muscle performance measure with a handheld or isokinetic dynamometer or electromyography, cervical or thoracic active ROM, active or latent trigger points, and posture) was carried out in non-athlete patients, understanding athlete's population as described by *Araújo & Scharhag (2016)*, diagnosed with one of the terms included under the overarching term of RCRSP. In addition, the studies had to compare the RCRSP group to an asymptomatic group (AG), and/or could also show correlations or regressions of the RCRSP group. In the results, the studies had to detail information regarding the comparative statistical data of the different variables related to the examination of the musculoskeletal dysfunction.

Finally, to be included in the meta-analysis, a minimum methodological quality was required (*Lo, Mertz & Loeb, 2014*), with at least six points for the cross-sectional studies and four points for case–control and cohort studies analyzed with Newcastle-Ottawa Scale (NOS). And, data of the analyzed variables had to be represented in at least three studies.

### Data extraction

Two independent reviewers conducted the entire selection process (DMH and IEG). The flow chart shows the process carried out by each reviewer to select the studies. First, duplicate studies were eliminated throughout the all databases analyzed. Identified titles and abstract were screened according to the described inclusion and exclusion criteria. When the title or abstract did not contain enough information for its exclusion; that study would progress to the next phase. In the next phase, the studies that overcame the previous phases were read in full text and those that fulfilled all the inclusion criteria of this review were selected. After the last screening phase, the whole group of this review met to analyze, one by one, each of the studies which the two independent reviewers did not agree in order to reach a consensus. Additionally, a whole group consensus reviewed, one by one, the reference lists of included studies to identify and include additional studies that met the criteria (records identified from citation searching).

## Quality assessment

The NOS was used to assess the methodological quality. The NOS for cohort studies was adapted for the methodological assessment of cross-sectional studies (*Lo, Mertz & Loeb, 2014*). It is an appropriate scale for reviews that include a large volume of studies because it is short and has moderate reliability (*Hootman et al., 2011*). The scale assesses three main aspects: the selection of the sample, the comparability between groups, and the outcome presentation, assigning a value of 5, 2, and 3 stars, respectively, to each of the three aspects. Therefore, the methodological quality was categorized as follows:

- Good: 3 or 4 stars in the selection domain, 1 or 2 stars in the comparability domain and 2 or 3 stars in the exposure domain.
- Moderate: 2 stars in the selection domain, 1 or 2 stars in the comparability domain and 2 or 3 stars in the exposure domain.
- Bad: 0 or 1 star in the selection domain or 0 stars in the comparability domain or 0 or 1 star in the exposure domain.

To avoid biases, two different experienced reviewers (DMH and AGM) performed the quality analysis independently. In cases of disagreement, the whole research group intervened in a consensual way to solve it. Inter-rater concordance was analyzed using Cohen's Kappa coefficient ($k > 0.7$ means a high level of agreement between the two evaluators, 0.5–0.7 means a moderate level, and <0.5 means a low level (*Cohen, 1960*).

## Data synthesis and meta-analysis

A meta-analysis was conducted using RStudio software version 2023.06.0-431 (*RStudio Team, 2023*), based on R software version 4.3.1 (*R Core Team, 2023*), and the "metafor" package (https://cran.r-project.org/web/packages/metafor/index.html) following Viechtbauer's guidelines (*Viechtbauer, 2010*). The inclusion criteria for studies in the meta-analysis were as follows: the articles provided detailed information on the statistical data of exposure factors; patients with RCRSP were compared to a control group of asymptomatic subjects; and the data for the analyzed variables were present in at least three studies.

A random-effects model with restricted maximum likelihood was used to pool different effect sizes due to the variability among interventions and comparisons, aiming to minimize the influence of heterogeneity among studies. This method is appropriate for continuous variables with a small number of studies (*Viechtbauer, 2016*).

Standardized mean differences (SMD) and 95% confidence intervals were calculated for the continuous variables to compare different study outcomes. The SMD's statistical significance was analyzed using Hedge's g to assess the possible overestimation of the true effect size in the population in small studies g. The interpretation of the estimated SMD was based on the study by *Hopkins et al. (2009)*, where an SMD of 4.0 represents an extremely large clinical effect, 2.0–4.0 a very large effect, 1.2–2.0 a large effect, 0.6–1.2 a moderate effect, 0.2–0.6 a small effect, and 0.0–0.2 an insignificant effect.

The degree of heterogeneity among the included studies was assessed using Cochranes' Q statistical test, with a $P$-value <0.05 considered significant, the inconsistency index (I2),
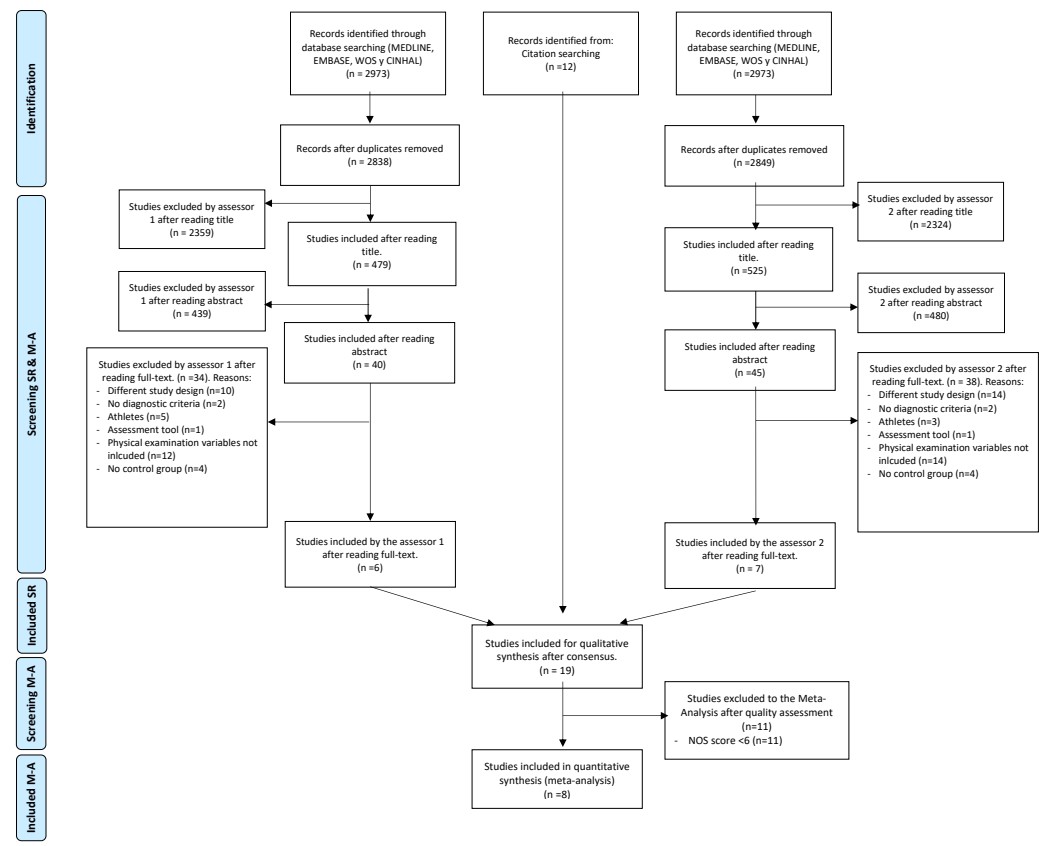

**Figure 1** **Flow chart of the study selection process.** NOS, Newcastle-Ottawa Scale.

and the value of the statistic Tau-squared (T2). T2, which shows the variance between studies explained by heterogeneity, was included to complement the interpretation of I2, which shows the proportion of variability between studies that is not due to sampling error, and not the degree of heterogeneity (*Rücker et al., 2008*). An I2 value >25% up to 50% is considered to represent a small proportion of heterogeneity that is not due to sampling error, I2 >50% up to 75% is considered moderate, and an I2 >75% is considered a large proportion of heterogeneity (*Huedo-Medina et al., 2006*; *Rücker et al., 2008*).

Funnel plot and Egger's regression test were employed to detect asymmetry due to possible publication biases (*Egger et al., 1997*; *Sterne & Egger, 2001*). Moreover, exclusion sensitivity analyses were used to evaluate the influence of each individual study in the different pooled estimations.

# RESULTS

## Flow of studies through the review

The electronic database search was conducted by two reviewers independently with high level of agreement ($k = 0.88$) and is shown in detail in Fig. 1.
*Study characteristics and demographics*

All the selected studies had a cross-sectional design which compared the RCRSP group with an AG, and no cohort or case-control studies were selected because they did not meet the inclusion criteria (*McClure, Michener & Karduna, 2006*; *Erol, Özçakar & Çeliker, 2008*; *Theisen et al., 2010*; *Land, Gordon & Watt, 2017*; *Kolber et al., 2017*; *Hunter et al., 2020*; *Ueda et al., 2020*; *Choi & Chung, 2023*).

*Hunter et al. (2020)* and *Ueda et al. (2020)* provided the diagnosis through imaging (MRI and ultrasound scan), two studies took into account physical and radiological examination (*Theisen et al., 2010*; *Choi & Chung, 2023*), whereas four studies diagnosed RCRSP patients based on physical examination, mainly orthopedic test (*McClure, Michener & Karduna, 2006*; *Erol, Özçakar & Çeliker, 2008*; *Land, Gordon & Watt, 2017*; *Kolber et al., 2017*).

A total of 604 participants were assessed (RCRSP = 298; AG = 306; 43.04% women, mean ages ranged from 67.0 to 26.4) (*McClure, Michener & Karduna, 2006*; *Erol, Özçakar & Çeliker, 2008*; *Theisen et al., 2010*; *Land, Gordon & Watt, 2017*; *Kolber et al., 2017*; *Hunter et al., 2020*; *Ueda et al., 2020*; *Choi & Chung, 2023*). One study only included men (*Kolber et al., 2017*). Seven studies matched both groups by age and gender (*McClure, Michener & Karduna, 2006*; *Erol, Özçakar & Çeliker, 2008*; *Theisen et al., 2010*; *Land, Gordon & Watt, 2017*; *Kolber et al., 2017*; *Hunter et al., 2020*; *Choi & Chung, 2023*), except (*Ueda et al., 2020*). Four studies showed data related to whether the dominant or non-dominant side was assessed (*McClure, Michener & Karduna, 2006*; *Erol, Özçakar & Çeliker, 2008*; *Theisen et al., 2010*; *Kolber et al., 2017*). Two studies provided information regarding time evolution of the patient's shoulder pain (8, 9 months) (*Erol, Özçakar & Çeliker, 2008*; *Ueda et al., 2020*), two studies mentioned that patients had had RCRSP at least for three months (*Theisen et al., 2010*; *Hunter et al., 2020*), between four weeks and twelve months (*Land, Gordon & Watt, 2017*) and the 64% more than three months (*McClure, Michener & Karduna, 2006*). Three studies showed the pain intensity on the examination day (4.44 ± 1.19 cm on a 10-cm visual analogue scale (VAS)) (*McClure, Michener & Karduna, 2006*; *Land, Gordon & Watt, 2017*; *Hunter et al., 2020*) and one study mentioned as inclusion criteria that shoulder pain must be rated at least 3 on the 10-point VAS (*Theisen et al., 2010*). Three studies assessed shoulder function *via* the Shoulder Pain Disability Index (SPADI) (38.80 ± 11.05) (*Erol, Özçakar & Çeliker, 2008*; *Land, Gordon & Watt, 2017*; *Hunter et al., 2020*), on the other hand, one study assessed shoulder function *via* the American Shoulder and Elbow Surgeons Self-reported Form (29.8/50) (*McClure, Michener & Karduna, 2006*).

## Quality assessment

The 8 studies showed an average methodological quality (8.75 ± 1.16 out of 10 possible points) and a range of 6 to 10 points (Table 1) (*McClure, Michener & Karduna, 2006*; *Erol, Özçakar & Çeliker, 2008*; *Theisen et al., 2010*; *Land, Gordon & Watt, 2017*; *Kolber et al., 2017*; *Hunter et al., 2020*; *Ueda et al., 2020*; *Choi & Chung, 2023*).

Two experienced reviewers independently performed the analysis, recording a high intertester reliability ($k = 0.93$). All disagreements between reviewers were then resolved by research group consensus.

Manoso-Hernando et al. (2024), *PeerJ*, DOI 10.7717/peerj.17604

**Table 1** Quality assessment of studies (NOS score of included studies).

| Authors (Year) | S1: Representativeness of the sample | S2: sample size | S3: Non-respondents | S4: Ascertainment of exposure | Ca: Study controls for most important factor | Cb: Study controls for additional factor | O1: Ax of outcome | O2: Statistical test | Total |
|---|---|---|---|---|---|---|---|---|---|
| Choi & Chung (2023) | ★ | ★ | ★ | ★★ | ★ | ★ | ★★ | | 9/10 stars |
| Ueda et al. (2020) | ★ | ★ | ★ | ★★ | ★ | ★ | ★★ | ★ | 10/10 stars |
| Hunter et al. (2020) | ★ | ★ | ★ | ★★ | ★ | | ★★ | ★ | 9/10 stars |
| Land, Gordon & Watt (2017) | ★ | ★ | ★ | ★★ | ★ | ★ | ★★ | ★ | 10/10 stars |
| Kolber et al. (2017) | ★ | ★ | ★ | ★★ | ★ | | ★★ | ★ | 9/10 stars |
| Theisen et al. (2010) | ★ | | ★ | ★★ | ★ | ★ | ★★ | ★ | 9/10 stars |
| Erol, Özçakar & Çeliker (2008) | ★ | | ★ | ★★ | ★ | | ★★ | | 7/10 stars |
| McClure, Michener & Karduna (2006) | ★ | | | ★★ | ★ | ★ | ★★ | | 7/10 stars |

**Notes.**
 NOS, Newcastle Ottawa Scale.

## Outcome measures

The characteristics of the sample, variables, measuring instruments, procedure, structure, and movement or orthopedic test results of studies are shown in Table 2. The following describes the characteristics of how each variable was measured.

### Meta-analysis of shoulder range of motion

A random-effects model was performed to analyse the heterogeneity of studies. The meta-analysis of these studies showed significantly less ROM with large effect size comparing the shoulder flexion ROM between participants with RCRSP and AG ($N = 308$, HG $= -1.30$, 95% CI [$-2.19$ to $-0.41$], $p < 0.01$) (Fig. 2) and shoulder IR ROM ($N = 355$, HG $= -1.32$, 95% CI [$-1.55$ to $-1.09$], $p < 0.01$) (Fig. 3) and significantly less ROM with moderate effect for ER ROM ($N = 308$, HG $= -1.16$, 95% CI [$-1.77$ to $-0.55$], $p < 0.01$) (Fig. 4).

Funnel plots to detect asymmetry due to possible publication biases are detailed in Files S1–S3. Furthermore, accordingly to Egger's test of asymmetry, the results suggested no significance evidence of publication bias for shoulder flexion ROM when comparing these groups ($t = 0.73$, $p = 0.54$) and IR ROM ($t = 1.28$, $p = 0.33$), but significance evidence of publication bias for shoulder ER ROM ($t = 11.79$, $p < 0.01$).

### Meta-analysis of shoulder strength

A random-effects model was performed to analyse the heterogeneity of studies. The meta-analysis of these studies showed no differences with large effect size comparing participants with RCRSP and AG for shoulder ABD strength ($N = 308$, HG $= -1.65$, 95% CI [$-3.88$–$0.59$], $p = 0.15$) (Fig. 5), and ER strength ($N = 346$, HG $= -1.54$, 95% CI [$-3.32$ to $-0.24$], $p < 0.09$) (Fig. 6), and IR strength differences with small effect size ($N = 346$, HG $= -0.31$, 95% CI [$-0.60$ to $-0.01$], $p < 0.05$) (Fig. 7).

Additionally, Egger's test showed no publication bias for shoulder ABD strength ($t = -1.63$, $p = 0.24$), shoulder ER strength ($t = -1.24$, $p = 0.30$) and shoulder IR strength ($t = 0.60$, $p = 0.59$). Funnel plots regarding shoulder strength are detailed in Files S4–S6.

### Meta-analysis of thoracic kyphosis

A random-effects model was performed. The meta-analysis of these three studies showed no differences and insignificant effect for the thoracic kyphosis in resting position in these groups ($N = 246$, HG $= 0.19$, 95% CI [$-0.29$–$0.67$], $p = 0.44$) (Fig. 8).

Furthermore, Egger's test showed no publication bias for the thoracic kyphosis in resting position ($t = 2.15$, $p = 0.06$). Funnel plot relative to this variable is detailed in File S7.

## DISCUSSION

In the recent years, there has been a change in the rotator cuff assessment paradigm as the systematic reviews and meta-analysis published up to date, were based on shoulder special test (*Hegedus et al., 2008*; *Hegedus et al., 2012*; *Gismervik et al., 2017*). This is the first meta-analysis that studies the differences in shoulder ROM, shoulder strength and thoracic kyphosis posture in patients with RCRSP compared to an AG. Only high quality studies were included. This meta-analysis found statistical differences between patients

Manoso-Hernando et al. (2024), *PeerJ*, DOI 10.7717/peerj.17604

**Table 2   Characteristics of each study group: Procedures, results, and differences in the physical examination between each study group.**

| Author (Year) (NOS) | Rotator cuff related shoulder pain group | Asymptomatic group | Measures | Tools | Movement/structure/ test | Outcomes/comparison |
|---|---|---|---|---|---|---|
| Cohen (1960) (9/10) | G1 (n = 55) M/F = 29/26 50.0 ± 4.2 years | G2 (n = 53) M/F = 30/23 50.3 ± 3.4 years | Shoulder ROM | Goniometer | Active Shoulder Flex-ion (degrees) | More subjects in G1 with reduced active shoulder flexion range than in G2 (p < 0.001) |
| | | | | | Active Shoulder ER (degrees) | More subjects in G1 with reduced shoulder ER range than in G2 (p < 0.001) |
| | | | | | Active Shoulder IR (degrees) | More subjects in G1 with reduced shoulder IR range than in G2 (p < 0.001) |
| | | | Shoulder strength | Isokinetic dynamometer | ER (Nm) | More subjects in G1 with reduced shoulder IR strength than in G2 (p < 0.001) |
| | | | | | IR (Nm) | NSD between groups (p = 0.3). |
| | | | | | ABD (Nm) | More subjects in G1 with reduced shoulder ABD strength than in G2 (p < 0.001) |
| Ueda et al. (2020) (10/10) | G1 (n = 32) M/F = 17/15 65.7 ± 6.5 years | G2 (n = 23) M/F = 15/8 67.0 ± 6.5 years | Shoulder ROM | | Active Shoulder Flex-ion (degrees) | More subjects in G1 with reduced active shoulder flexion range than in G2 (p < 0.001) |
| | | | | | Active Shoulder ER (degrees) | More subjects in G1 with reduced shoulder ER range than in G2 (p = 0.03) |
| | | | Shoulder strength | Isokinetic dynamometer | ER (Nm/Kg) | NSD between groups (p = 0.06). |
| | | | | | IR (Nm/Kg) | More subjects in G1 with reduced shoulder IR strength than in G2 (p < 0.001) |
| | | | | | ABD (Nm/Kg) | More subjects in G1 with reduced shoulder ABD strength than in G2 (p < 0.001) |
| Hunter et al. (2020) (9/10) | G1 (n = 39) M/F = 20/19 57.1 ± 11.1 years | G2 (n = 39) M/F = 20/19 55.7 ± 10.6 years | Thoracic kyphosis | X-ray | Modified Cobb's angle (degrees) | More subjects in G2 with reduced modified Cobb angle than in G1 (p = 0.004) |
| Land, Gordon & Watt (2017) (10/10) | G1 (n = 51) M/F = 28/23 51.2 ± 5.71 years | G2 (n = 51) M/F = 28/23 50.8 ± 4.6 years | Shoulder ROM | Plastic goniometer | Passive shoulder IR (degrees) | More subjects in G1 with reduced passive shoulder IR range than in G2 (p < 0.001) |
| Kolber et al. (2017) (9/10) | G1 (n = 24) M/F = 24/0 28.5 ± 9.1 years | G2 (n = 31) M/F = 31/0 26.4 ± 3.1 years | Shoulder ROM | Goniometer | Active Shoulder Flex-ion | NSD between groups (p = 0.37) |
| | | | | | Active Shoulder ER | More subjects in G1 with reduced shoulder ER range than in G2 (p < 0.001) |
| | | | | | Active Shoulder IR | More subjects in G1 with reduced shoulder IR range than in G2 (p = 0.016) |
| | | | Shoulder strength | Handheld dynamometer | ER (Kg) | More subjects in G1 with reduced shoulder ER strength than in G2 (p < 0.001) |
| | | | | | IR (Kg) | NSD between groups (p = 0.38) |
| | | | | | ABD (Kg) | NSD between groups (p = 0.92) |
| Theisen et al. (2010) (9/10) | G1 (n = 39) M/F = 23/16 56.6 ± 38-77 years | G2 (n = 39) M/F = 23/16 56.1 ± 38-79 years | Static Thoracic kyphosis | Ultrasound topometrics | Static kyphosis | NSD between groups (p > 0.05) |

Manoso-Hernando et al. (2024), *PeerJ*, DOI 10.7717/peerj.17604

Manoso-Hernando et al. (2024), *PeerJ*, DOI 10.7717/peerj.17604

**Table 2** (*continued*)

| Author (Year) (NOS) | Rotator cuff related shoulder pain group | Asymptomatic group | Measures | Tools | Movement/structure/ test | Outcomes/comparison |
|---|---|---|---|---|---|---|
| *Erol, Özçakar & Çeliker (2008)* (7/10) | G1 (*n* = 13) *M/F* = 3/10 37.8 ± 9.4 years | G2 (*n* = 25) *M/F* = 5/20 37.1 ± 9 years | Shoulder strength | Isokinetic dynamometer | ER (Nm) | NSD between groups (*p* = 0.56) |
| | | | | | IR (Nm) | NSD between groups (*p* = 0.31) |
| *McClure, Michener & Karduna (2006)* (7/10) | G1 (*n* = 45) *M/F* = 24/21 45.2 ± 12.8 years | G2 (*n* = 45) *M/F* = 24/21 43.6 ± 12.4 years | Shoulder ROM | Goniometer | Active Flexion (degrees) | More subjects in G1 with reduced active shoulder flexion than in G2 (*p* < 0.001) |
| | | | | | Active ER (degrees) | More subjects in G1 with reduced active shoulder ER than in G2 (*p* < 0.001) |
| | | | | | Active IR (degrees) | More subjects in G1 with reduced active shoulder IR than in G2 (*p* < 0.001) |
| | | | Shoulder strength | Handheld dynamometer | ER (Kg) | More subjects in G1 with reduced shoulder ER strength than in G2 (*p* < 0.001) |
| | | | | | IR (Kg) | NSD between groups (*p* = 0.11) |
| | | | | | ABD (Kg) | More subjects in G1 with reduced shoulder ABD strength than in G2 (*p* < 0.001) |
| | | | Thoracic Kyphosis | Gravity inclinometer | Thoracic spine inclination (degrees) | NSD between groups (*p* = 0.415) |

**Notes.**

ROM, Range of motion; ER, External rotation; IR, Internal rotation; ABD, Abduction; NSD, No statistical differences; Nm/Kg, Newton metre per kilogram; Kg, Kilograms; Nm, Newton meter.
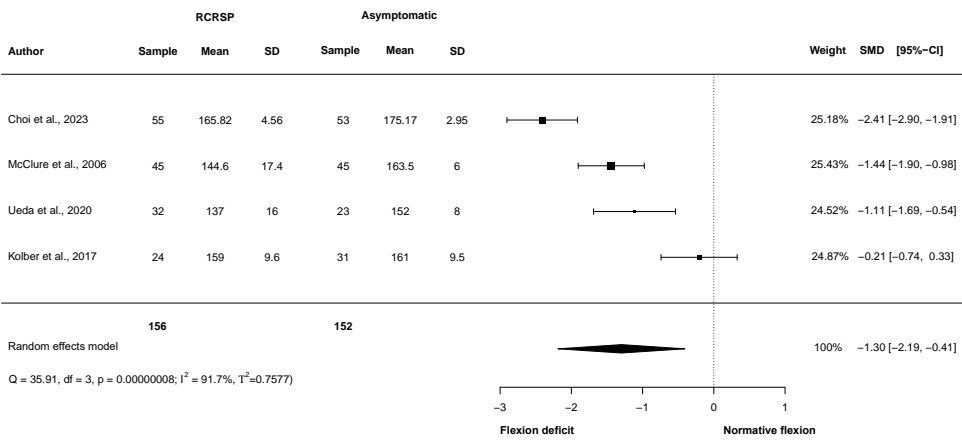

**Figure 2** **Synthesis forest plots: Shoulder flexion ROM.** RCRSP, Rotator cuff related shoulder pain; SD, Standard Deviation; SMD, Standarized Mean Difference; CI, Confidence Interval; Q, Cochran's Q; df, Degrees of freedom; *p*, *p*-value of Cochran's Q; I2, Index of inconsistency; T2, Tau squared.

**Figure 3** **Synthesis forest plots: Shoulder IR ROM.** RCRSP, Rotator cuff related shoulder pain; SD, Standard Deviation; SMD, Standarized Mean Difference; CI, Confidence Interval; Q, Cochran's Q; df, Degrees of freedom; *p*; *p*-value of Cochran's Q; I2, Index of inconsistency; T2, Tau squared.

with RCRSP and an AG, finding a diminished shoulder flexion, ER and IR ROM and less IR strength.

## Range of motion

Active ROM is one of the most important variables in joint function and is also considered one of the variables that best explain patient's quality of life (*Murata et al., 2017*). Differences and large effect sizes were found in shoulder flexion, ER and IR, indicating less ROM in patients with RCRSP compared to an AG. The ROM was measured in all the

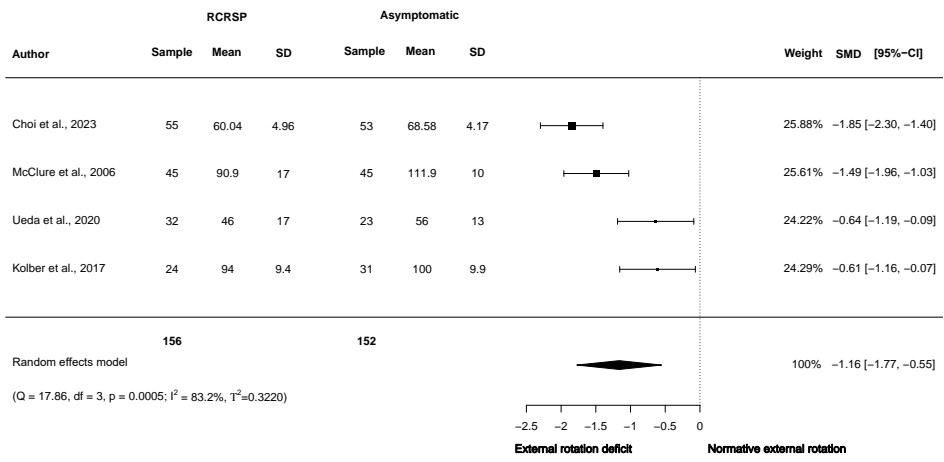

**Figure 4** **Synthesis forest plots: Shoulder ER ROM.** RCRSP, Rotator cuff related shoulder pain; SD, Standard Deviation; SMD, Standarized Mean Difference; CI, Confidence Interval; Q, Cochran's Q; df, Degrees of freedom; *p*, *p*-value of Cochran's Q; I2, Index of inconsistency; T2, Tau squared.

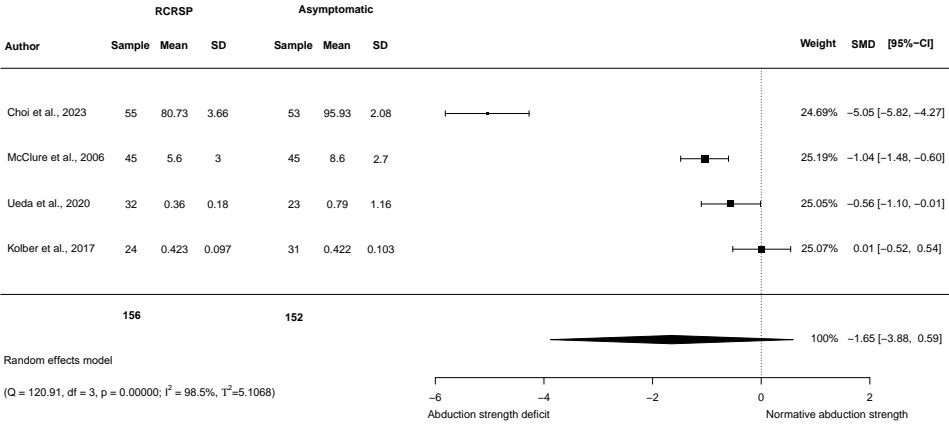

**Figure 5** **Synthesis forest plots: Shoulder ABD strength.** RCRSP, Rotator cuff related shoulder pain; SD, Standard Deviation; SMD, Standarized Mean Difference; CI, Confidence Interval; Q, Cochran's Q; df, Degrees of freedom; *p*, *p*-value of Cochran's Q; I2, Index of inconsistency; T2, Tau squared.

studies included in this meta-analysis using a goniometer (*McClure, Michener & Karduna, 2006*; *Erol, Özçakar & Çeliker, 2008*; *Theisen et al., 2010*; *Land, Gordon & Watt, 2017*; *Kolber et al., 2017*; *Hunter et al., 2020*; *Choi & Chung, 2023*), however, *Ueda et al. (2020)*, did not specify the instrument used.

It must be mentioned that *Land, Gordon & Watt (2017)* assessed the passive IR whereas Choi, Kolber and McClure examined the active IR (*McClure, Michener & Karduna, 2006*; *Kolber et al., 2017*; *Choi & Chung, 2023*), therefore the results should be taken with caution. There is also no consensus in terms of patient position throughout IR ROM assessment, whereas *Kolber et al. (2017)* examined this variable in prone, two studies assessed it in supine with shoulder in 90° of ABD *McClure, Michener & Karduna (2006)*,
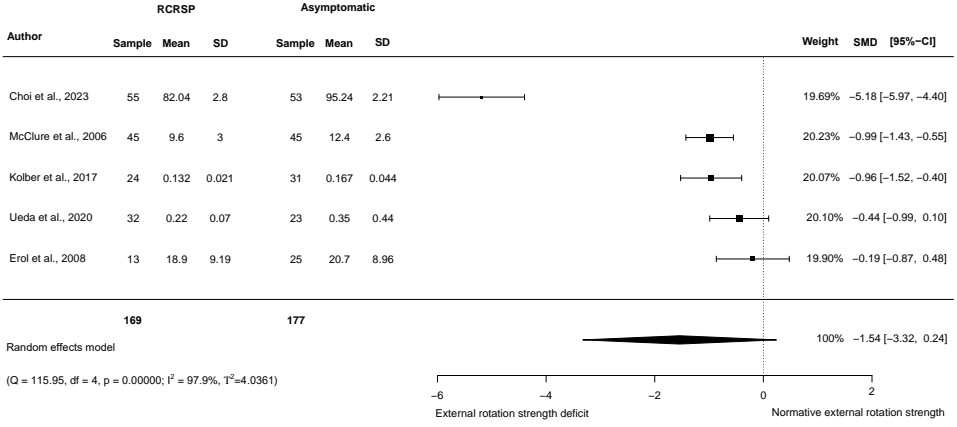

**Figure 6** **Synthesis forest plots: ER shoulder strength.** RCRSP, Rotator cuff related shoulder pain; SD, Standard Deviation; SMD, Standarized Mean Difference; CI, Confidence Interval; Q, Cochran's Q; df, Degrees of freedom; p, p-value of Cochran's Q; I2, Index of inconsistency; T2, Tau squared.

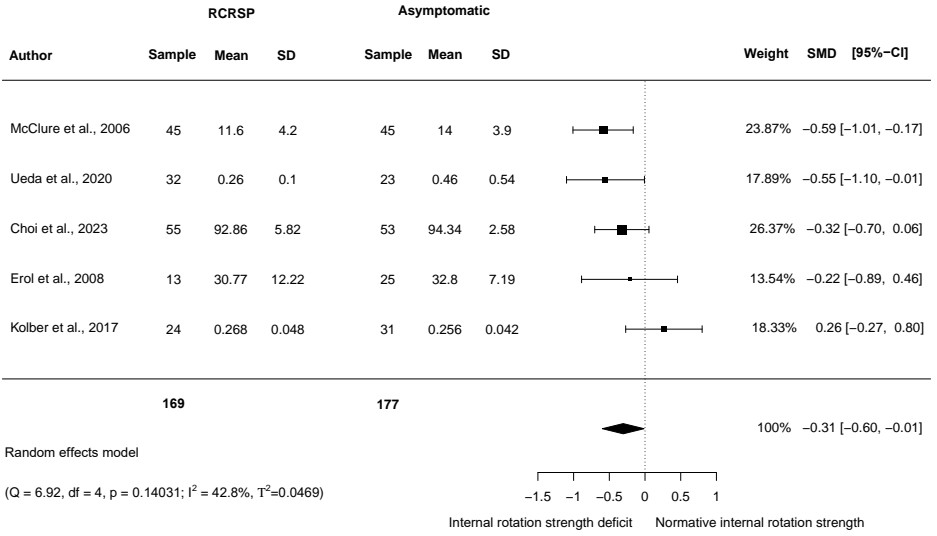

**Figure 7** **Synthesis forest plots: IR shoulder strength.** RCRSP, Rotator cuff related shoulder pain; SD, Standard Deviation; SMD, Standarized Mean Difference; CI, Confidence Interval; Q, Cochran's Q; df, Degrees of freedom; p, p-value of Cochran's Q; I2, Index of inconsistency; T2, Tau squared.

*Choi & Chung (2023)* and *Land, Gordon & Watt (2017)* examined this variable with the patient in supine with the humerus in 90° of ABD and the tester palpating the scapula while passively internally rotating the shoulder, however, it seems clear that this variability during the assessment method do not influence the results as in all cases IR ROM was diminished. Regarding ER ROM, all studies measured active ROM. Two studies measured ER in supine with shoulder in 90° of shoulder ABD *McClure, Michener & Karduna (2006)*, *Choi & Chung (2023)*, *Kolber et al. (2017)* also evaluated it in supine but did not specify the

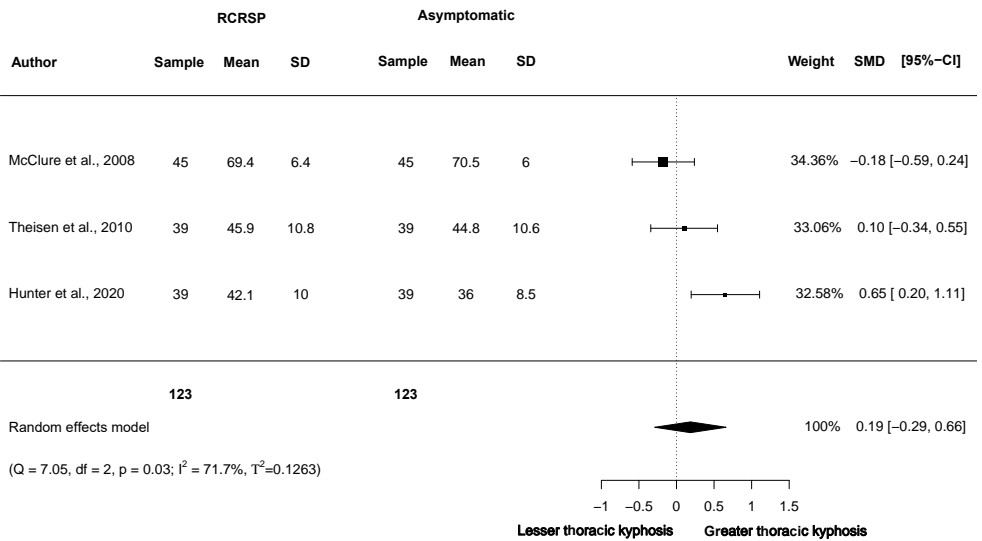

**Figure 8** **Synthesis forest plots: Thoracic static kyphosis.** RCRSP, Rotator cuff related shoulder pain; SD, Standard Deviation; SMD, Standarized Mean Difference; CI, Confidence Interval; Q, Cochran's Q; df, Degrees of freedom; *p*, *p*-value of Cochran's Q; I2, Index of inconsistency; T2, Tau squared.

arm position and another study did not mention neither the patent's nor the arm position (*Ueda et al., 2020*).

The fact that RCRSP patients presented less ROM in shoulder flexion, ER and IR might be associated with the presence of pain during the movement execution, when the rotator cuff has to deal with different loads (*Cools et al., 2020*), however none of the studies included in this meta-analysis took pain into account during the movement execution.

## Strength

In the current literature, the association between RCRSP and strength is remarkable, as the main goal in today's practice is to enhance tendon loading capacity (*Littlewood et al., 2012*; *Littlewood, Malliaras & Chance-Larsen, 2015*; *Malliaras et al., 2020*; *Dubé et al., 2020*). The evidence presented in this meta-analysis showed differences with small effect size for a diminished shoulder IR strength in patients with RCSPS but no differences in shoulder ABD and ER.

However, three studies showed statistical differences in shoulder ABD strength when comparing patients with RCRSP to an AG (*McClure, Michener & Karduna, 2006*; *Ueda et al., 2020*; *Choi & Chung, 2023*), whilst only Kolber et al. found no statistical differences (*Kolber et al., 2017*). This contrast might be explained as the study sample in the latter were weight training participants, hence this discrepancy may be justified as exercise is considered to be both essential and at the forefront of the management of RCRSP (*Powell et al., 2022*), this could explain why these patients did not present shoulder ABD strength deficit. Also, the age mean in this study was 27 compared to *McClure, Michener & Karduna (2006)*, *Ueda et al. (2020)* and *Choi & Chung (2023)*, where the age mean was 50.5, 45.2 and 65.7, respectively, as shoulder pain generally increases past 50 (*Leong et al., 2019*; *Hodgetts*

*et al., 2021*) this fact might also increase the likelihood of statistical differences between studies.

Equally, three studies found statistical differences in ER shoulder strength (*McClure, Michener & Karduna, 2006*; *Ueda et al., 2020*; *Choi & Chung, 2023*), whereas *Kolber et al. (2017)* and *Erol, Özçakar & Çeliker (2008)* did not find statistical differences between both groups. In our opinion, these statistical discrepancies may be explained as *Erol, Özçakar & Çeliker (2008)* and *Kolber et al. (2017)* utilized a handheld dynamometer as the assessment tool, whereas *McClure, Michener & Karduna (2006)*, *Ueda et al. (2020)*, and *Choi & Chung (2023)* used an isokinetic dynamometer, assessing the isometric peak force and the peak torque at 60° /s, however, in literature, no significant differences have been found between both instruments (*Stark et al., 2011*).

Exclusion sensitivity analyses showed the influence of *Choi & Chung (2023)* in the ER strength pooled estimation, as its exclusion significantly influenced the SMD (−0.7), 95% IC (−1.06 to −0.32), $p$-valor ($p < 0,01$) and I2 (45.02%).

The reason why both shoulder ABD and ER strength had no statistical differences, even though the majority of the studies included showed statistical differences, might be explained due to the substantial heterogeneity found in these two variables.

Whereas it seems clear that all patients with RCRSP presented less strength in IR, the findings related to shoulder ABD and ER shoulder strength are more inconclusive.

## Thoracic spine

This meta-analysis has not found statistical differences in the thoracic spine static kyphosis between RCRSP patients and an AG.

Although previous studies found a relationship between a large thoracic kyphosis and reduced shoulder flexion in healthy population (*Crawford & Jull, 1993*) and another study has demonstrated that approximately 15° and 9° of thoracic extension mobility is required for full bilateral and unilateral shoulder flexion (*Stewart et al., 1995*), this meta-analysis concurs with the systematic review published by *Barrett et al. (2016)*, in which it was reported that thoracic kyphosis may not be an important contributor to the development of shoulder pain.

## Clinical implications

The methods previously proposed in the literature to assess patients with RCRSP, based on special orthopedic tests, have been discarded in recent publications (*Hughes, Taylor & Green, 2008*; *Lewis, 2009b*; *Salamh & Lewis, 2020*). Thus, this opens the possibility of a different method of assessment where shoulder ROM, muscle strength and the examination of the thoracic spine seems to be an accepted method to rule in or rule out rotator cuff pathology throughout the physical examination, as it has been proposed recently (*Littlewood et al., 2019*; *Salamh & Lewis, 2020*). The present study highlights the importance of considering shoulder ROM, especially shoulder flexion, ER and IR in the examination process or through the Constant Score as shoulder ROM is often collected in clinical practice employing this scale and IR strength, as statistical differences have been reported in this meta-analysis. Thoracic static kyphosis seems not to be a significant

variable to take into account in RCRSP patients, however, we propose for further studies the assessment of thoracic ROM as it might overcome the limitations of simply assessing static position.

Nonetheless, future studies of high methodological quality should also consider the analysis of other variables of shoulder ROM, as they were not analyzed because they were not included in this meta-analysis. Furthermore, we also propose to clarify in future studies if shoulder ROM impairments are related to pain, joint stiffness or a different cause. In addition, more research is needed to enhance the understanding of the role of shoulder and scapular kinematics in RCRSP etiology, as it has been proposed in the literature (*Lawrence, Braman & Ludewig, 2020*). This can lead to further identify the cause of shoulder pain and how rehabilitation can aim to address this. In this sense, to identify subjects at risk for developing shoulder problems who may still be asymptomatic but already present some signs of tendon degeneration seems a crucial factor to prevent shoulder pain (*Lawrence et al., 2024*).

The lack of a greater volume of high-quality studies and the substantial heterogeneity make it difficult to establish a clear conclusion that associates RCRSP with shoulder ABD and ER strength impairments and we recommend to take these results with caution, hence the authors of this meta-analysis strongly recommend further studies that will provide more accurate data in this field and perhaps that future research should consider the possibility of stratifying RCRSP patients according to their clinical presentation.

## Limitations

This meta-analysis has several limitations. First, conclusions based on the meta-analysis results are limited due to heterogeneity and the small number of included studies. Second, *Erol, Özçakar & Çeliker (2008)* provided median and interquartile range and therefore mean and standard deviation were calculated from these data. Third, some variables had been evaluated using different assessment methods such as the make and break test with a handheld dynamometer or the isometric peak torque and peak torque at 60° /s with an isokinetic dynamometer in shoulder strength (*McClure, Michener & Karduna, 2006*; *Erol, Özçakar & Çeliker, 2008*; *Kolber et al., 2017*; *Ueda et al., 2020*; *Choi & Chung, 2023*). Other study did not mention the examination tool throughout the shoulder ROM assessment (*Ueda et al., 2020*). On the other hand, regarding IR ROM, it must also be mentioned that *Land, Gordon & Watt (2017)* assessed the passive IR, whereas *Choi & Chung (2023)*, *Kolber et al. (2017)* and *McClure, Michener & Karduna (2006)* examined the active IR. The ideal would be the ability to unify both the units of measurement and the evaluation instruments to facilitate the interpretation of the data obtained. Lastly, RCRSP is an umbrella term that encompasses different terminology such as subacromial impingement syndrome, rotator cuff tendinopathy, bursitis and symptomatic partial and full thickness rotator cuff tears (*Lewis, 2016*), in this study, seven articles included patients diagnosed with subacromial impingement syndrome (*McClure, Michener & Karduna, 2006*; *Erol, Özçakar & Çeliker, 2008*; *Theisen et al., 2010*; *Land, Gordon & Watt, 2017*; *Kolber et al., 2017*; *Hunter et al., 2020*; *Choi & Chung, 2023*) and one study included patients with rotator cuff tears (*Ueda*

*et al., 2020*), therefore, the results of this study might not be generalizable to the whole spectrum of RCRSP patients.

In addition, although the search covered all observational designs, only cross-sectional studies were found that met our selection criteria. This scenario limits the conclusion of the meta-analysis, as it does not allow establishing a causal relationship between the study variables.

# CONCLUSIONS

This meta-analysis concludes that patients with RCRSP present less ROM in shoulder flexion, ER and IR and manifest less strength in shoulder IR compared to an AG. No differences were found in shoulder ABD and ER strength and static thoracic kyphosis between groups. These findings must be interpreted with caution due to the presence of heterogeneity and the limited number of included studies. Nonetheless, current findings could be applied in shoulder physical examination, as these results highlight the importance of assessing shoulder ROM and IR strength in patients with suspected RCRSP apart from shoulder ABD and ER strength as it was previously proposed in the literature.

# ACKNOWLEDGEMENTS

We would like to thank Keryn Legg for her help in editing/translating this article.

## Funding
The authors received no funding for this work.

## Competing Interests
Alfonso Gil-Martínez is an Academic Editor for PeerJ.

## Author Contributions
- Daniel Manoso-Hernando conceived and designed the experiments, performed the experiments, prepared figures and/or tables, authored or reviewed drafts of the article, and approved the final draft.
- Javier Bailón-Cerezo conceived and designed the experiments, performed the experiments, authored or reviewed drafts of the article, and approved the final draft.
- Santiago Angulo-Díaz-Parreño analyzed the data, prepared figures and/or tables, authored or reviewed drafts of the article, and approved the final draft.
- Álvaro Reina-Varona analyzed the data, prepared figures and/or tables, authored or reviewed drafts of the article, and approved the final draft.
- Ignacio Elizagaray-García conceived and designed the experiments, performed the experiments, authored or reviewed drafts of the article, and approved the final draft.
- Alfonso Gil-Martínez conceived and designed the experiments, performed the experiments, authored or reviewed drafts of the article, and approved the final draft.

## Data Availability

The raw data is available in the Supplemental Files.

## Supplemental Information

Supplemental information for this article can be found online at http://dx.doi.org/10.7717/peerj.17604#supplemental-information.

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
