# Peer review of "Shoulder mobility and strength impairments in patients with rotator cuff related shoulder pain: a systematic review and meta analysis"

_PeerJ, doi:10.7717/peerj.17604_

## Round 0.1 · original submission · Major Revisions

· Academic Editor

Major Revisions

Please be more specific within your methodological approach. Which studies were excluded and why. And which populations (e.g., athletes) were excluded and why.

Reviewer 1 ·

Basic reporting

Overall, the study is well written and easy to follow. All necessary data are provided.
Line 26 needs rewording, as it is true that clinical tests are not able to precisely identify the source of pain, these tests are still performed in clinical practice.
Sentence at line 144 could be removed. This information belongs to data synthesis.
Some words have a wrong ‘-‘ or ‘¬’ in the middle (e.g. filters at line 112, cross-sectional at line 142, cohort at line 143)
Line 127, the period at end of the sentence is missing.
Lines 197: Hedges’ g
Line 397: asymptomatic group should be replaced with its abbreviation introduced before.
Figure legends are incomplete. Abbreviation used in the figures should be added to the figure legends.
Reference to supplemental files should be added to the main text.

Experimental design

The aim and research questions are well defined and meaningful.
The search and screening process performed in this study is not very common. Usually, the search and duplicate removal is done just by one person. Why was the search performed by two reviewers independently? Why do the two Reviewers have not the same number of studies after duplicate removal (Fig 1)? Did reviewer 1 check also the additional articles?
Related to line 129, was forward citation search performed? If not, why?
What was the rationale for the first screening filter (line 150)? What criteria were used to exclude studies just based on the title?

Validity of the findings

Statistical analyses are appropriate. However, only few studies have been included in this systematic review and results might therefore not be generalizable to the whole spectrum of RCRSP conditions. Of the 8 studies included, 7 had patients with subacromial impingement syndrome, and one had patients with rotator cuff tears. Depending on the severity of the impingement syndrome, there might be rotator cuff tears, but rotator cuff tears do not necessarily imply subacromial impingement.
At line 347 it is mentioned the lack of consensus for the assessment of the ROM for internal rotation, what about external rotation?
Lines 413-414 shoulder range of motion is often already collected in clinical practice with the Constant Score, which covers also pain, activities of daily live and strength at 90° abduction.
In the conclusions, the main results are summarised again, but it would be important to say what the relevance of these results for clinicians is.

Reviewer 2 ·

Basic reporting

While the publication clearly discussed the literature on the concept of rotator cuff related shoulder pain and implications on shoulder outcomes (range of motion and strength). I am missing literature on the background and etiology of rotator cuff related shoulder pain as well as the impact of scapular kinematic. What is the cause of the problem and how to address this should be discussed and considered as well. Furthermore, the critical concern that many persons who are asymptomatic do present with pathology has to be discussed. e.g. pointing towards the need of early identification where tendons can still positively adapt to changes of loading (e.g. Ref. Lawrence RL, Moutzouros V, Bey MJ. Asymptomatic Rotator Cuff Tears. JBJS Rev. 2019 Jun;7(6):e9. doi: 10.2106/JBJS.RVW.18.00149. PMID: 31246863; PMCID: PMC7026731. as well as Lawrence RL, Soliman SB, Dalbøge A, Lohse K, Bey MJ. Investigating the multifactorial etiology of supraspinatus tendon tears. J Orthop Res. 2023 Oct 10. doi: 10.1002/jor.25699. Epub ahead of print. PMID: 37814893.)
The results section is rather extensive, information could be provided more concisely and clearly by avoiding repetition of information that is also in the tables.

Experimental design

The inclusion of thoracic kyphosis should be more introduced in the introduction.
While the presented findings are valuable to understand how range of motion and shoulder strength may be altered in symptomatic vs asymptomatic persons, clear limitations should be discussed related to the lack of information on scapular kinematics, identifying the cause of shoulder problems and how rehabilitation can aim to address this (e.g. optimize shoulder loading, restore healthy scapular motion), what about asympatomatic patients with shoulder pathology, the limitations of current assessment methods, could there be other assessment methods more sensitive to identify subjects at risk for developing shoulder problems who may still be asymptomatic but already present some signs of tendon degeneration?

Validity of the findings

All comments above impact the validity of the findings

Additional comments

The first sentence in the abstract is confusing to me. In general, it is unclear to me what is meant with '.... has been recently discarded..'? Objective of the abstract should state info related to the systematic review and meta analysis.
In the inclusion it is stated that athletes are excluded? but in the discussion and results there is a comment that some of the subjects did strength training? How were non-athlete patients defined exactly and why would such studies have been excluded?

---

## Round 0.2 · accepted · Accept

· Academic Editor

Accept

The authors carefully responded to each of the reviewers' comments and clarity of the manuscript further improved.

Reviewer 1 ·

Basic reporting

No comment

Experimental design

No comment

Validity of the findings

No comment

Additional comments

Thank you for revising the manuscript according to the previous comments. All necessary changes have been made.

Reviewer 2 ·

Basic reporting

Line 96: I believe there may be some spelling mistakes in this sentence, the 'not' should be removed here right? Furthermore, instead of none systematic review would be 'no systematic review'?

Experimental design

no comment

Validity of the findings

no comment

Additional comments

The authors carefully responded to each of the reviewers' comments and clarity of the manuscript further improved. The extensive and detailed response is highly appreciated.